# Effect of Three Months Pilates Training on Balance and Fall Risk in Older Women

**DOI:** 10.3390/ijerph18073663

**Published:** 2021-04-01

**Authors:** Małgorzata Długosz-Boś, Katarzyna Filar-Mierzwa, Robert Stawarz, Anna Ścisłowska-Czarnecka, Agnieszka Jankowicz-Szymańska, Aneta Bac

**Affiliations:** 1Sports and Recreation Center, Pedagogical University of Krakow, 30-084 Kraków, Poland; malgorzata.dlugosz-bos@up.krakow.pl; 2Institute of Applied Sciences, Faculty of Motor Rehabilitation, Bronislaw Czech University of Physical Education in Krakow, 31-571 Krakow, Poland; katarzyna.filar@awf.krakow.pl (K.F.-M.); anna.scislowska@awf.krakow.pl (A.Ś.-C.); 3Institute of Biology, Pedagogical University of Krakow, 30-084 Kraków, Poland; robert.stawarz@up.krakow.pl; 4Faculty of Health Science, University of Applied Science in Tarnów, 33-100 Tarnów, Poland; jankowiczszymanska@pwsztar.edu.pl

**Keywords:** balance, Pilates, falls, seniors, exercises

## Abstract

This study assessed the effect of Pilates exercises on balance and fall risk in older women. Participants comprised 50 older women aged over 60 years, divided randomly into two groups: the experimental group (*n* = 30), which took part in Pilates sessions two times per week for three months, and the control group (*n* = 20). The control group did not participate in such sessions but also did not participate in any other rehabilitation programs or additional physical activity except everyday activities. Before and after the training cycle, all women underwent an assessment using Timed Up and Go (TUG), the One Leg Stance Test (OLST), a test performed on a Freestep baropodometric platform, and the tests performed on a Biosway platform. After the training, significantly decreased values of the surface of the ellipse (*p* = 0.0037) and mean values of velocity (*p* = 0.0262) for the right foot in the experimental group were observed. The Limits of Stability (LoS) test (*p* = 0.005) and the Modified Clinical Test of Sensory Interaction on Balance (m-CTSIB) performed on an unstable surface with eyes closed (*p* = 0.0409) indicated statistically significant changes in the experimental group. None of the above changes were statistically significant in relation to the control group. Pilates training affected the participants’ balance by improving LOS and reducing fall risk.

## 1. Introduction

Old age involves many involutionary changes in the body that lead to considerable limitations and dysfunctions and reduced psychomotor activity [1]. Involutionary changes related to aging affect all primary systems of the body, including anatomical and functional systems [2]. As a result, the effectiveness of postural and motor systems responsible for stability decreases with age [3]. Maintaining functional fitness and, consequently, independence seems to be the key factor affecting the psychomotor condition of seniors [2]. Balance disorders and falls are two significant issues that may impact the functioning of an aging population [3].

The main task of the balance system is to keep the body’s center of mass at equilibrium, both at rest and during activity, by generating the appropriate responses to control body posture [4]. These responses are based on information coming from the proprioceptors, eyes and vestibular system [4]. However, sensory organs undergo structural and functional changes with age, which leads to balance disorders [4].

With age, proprioception diminishes, the number of cutaneous receptors in the legs decreases and the threshold for vibrational stimuli changes, all of which lead to lower postural stability in seniors [5]. Furthermore, neural conductivity diminishes, leading to longer reaction times, and the integration of sensory and motor responses becomes impaired [6,7]. Sometimes joint degeneration disorders also appear which affect body posture and in this way may contribute to postural instability [6]. All these processes impact the ability to maintain balance [7]. In old age, the muscular system loses mass and strength, causing difficulties with maintaining balance [8], which is mainly connected with core muscles which are responsible for trunk stability. From a practical perspective, the core muscles are the center of the body where most kinetic chains transfer forces to the extremities. Many types of exercises are recommended to improve core muscle strength and stability [9].

Balance disorders cause falls, which in turn make seniors more reliant on support from other people [10]. Women even fall three times more often than men and are hospitalized five times more often due to falls [11,12].

Many studies confirm that an appropriate volume of physical activity can improve visual perception and psychomotor coordination, and strengthen the muscles, all of which improve balance and reduce fall risk [2,13,14]. One of the forms of physical activity suggested for seniors is Pilates. The subject literature contains a growing number of studies that confirm the effectiveness of Pilates in improving various aspects of health [15]. Curi et al. [16] assessed the effect of Pilates on satisfaction, happiness and the functional performance of the body, including the strength and elasticity of the upper and lower body, respiratory function and dynamic balance. The results confirmed the effectiveness of Pilates in all assessed indicators. Furthermore, another study demonstrated that Pilates decreased tension and pain in the spine [17], strengthened weak muscles, stretched tight muscles [18] and improved overall functional fitness [19]. In the elderly, Pilates also affects the strength and mass of lower limb muscles which helps to prevent loss of balance [16,20,21,22].

The many benefits of Pilates exercises have motivated the authors of this paper to conduct a study in order to provide a broader perspective on balance disorders, which are one of the primary difficulties experienced by seniors. The authors decided to use Pilates exercises in the research because Pilates is a safe form of movement for older people. This type of training may be an alternative to other forms of exercise, which are too difficult or impossible to perform in older people, like fitness classes or gym training. The calm nature of Pilates exercises has a beneficial effect on the seniors mental state [23]. Regular Pilates practice can not only reduce the risk of depression, but also improve the quality of life and well-being of older people [24,25,26]. Moreover, Pilates, the main purpose of which is to strengthen the trunk muscles, improves the general body fitness and has a positive effect on moving and performing everyday activities [27]. It is very important in maintaining the independence of seniors. Consequently, the aim of this study was to assess the effect of Pilates exercises on balance and fall risk in older women.

## 2. Materials and Methods

### 2.1. Materials

Study participants comprised 50 older women aged over 60 years (see Table 1). The participants were divided randomly into two groups: an experimental (30 women) and a control (20 women) (see Figure 1). Qualification was based on simple randomization (a coin toss) and was carried out by the main author. The experimental group took part in Pilates sessions two times per week for three months. The control group did not participate in such sessions but also did not participate in any other rehabilitation programs or additional physical activity except everyday activities. All participants declared that they would not take part in other sports activities during the project.

The inclusion criteria for the study were as follows: age over 60 years, consent for participation in the study, no contraindications to physical activity and (experimental group only) a declaration of consistent participation in the Pilates sessions. The exclusion criteria were as follows: motor disability that prevented independent activity; neurological diseases that caused balance disorders; poor mental state (e.g., depression); physical conditions that prevented participation, e.g., severe respiratory or circulatory deficiency; and contraindications to physical activity.

The research project was approved by the Bioethics Committee as Study No. 189/KBL/OIL/2017. Furthermore, all women who applied to participate in the study were informed about the medical considerations and provided informed consent for participation.

### 2.2. Methods

Body height was measured using a Martin anthropometer (Seritex, New York, NY, USA) to an accuracy of 1 mm. Body mass was measured using a Tanita scale (Tanita Corporation, Tokyo, Japan) to an accuracy of 100 g. These variables were measured once.

In addition, regarding the balance and fall risk, these were measured twice. In this regard, Timed Up and Go (TUG) was used to assess functional fitness and fall risk [28]. The score was based on the time it took a participant to complete the test. The test consisted of several elements. The participant started by sitting in a chair. After the supervisor called out “Go!”, the participant got up from the chair, walked three meters, made a half turn (180°) and went back to the chair and sat down. The timer was stopped when the participant sat down again. A score of 10 s or below was considered the boundary of good functional fitness. A score above 13.5 s indicated an increased risk of balance disorders and falls [29].

The One Leg Stance Test assessed the ability to maintain balance on a decreased plane of support. Transferring one’s body weight onto a single leg is a natural everyday action, for instance when turning, going up and down stairs, stepping into a bathtub or shower and, most importantly, walking, in which the one-leg support phase takes up as much as 80% of the entire cycle. Unfortunately, the subject literature is in disagreement with regard to the testing procedure, which makes referencing similar studies difficult [30]. This study, in order to obtain comprehensive results, assessed the participants’ ability to stand on each leg, with focus on the dominant leg. The timer started when a participant lifted her leg up to the height of least half of her calf. One test was performed for each leg. The dominant leg was assumed to be the one the participant chose intuitively in the first assessment. The maximum time spent standing on one leg was assumed to be 30 s, because according to Hurvitza et al. [31], maintaining balance on one leg for less than 30 s correlates strongly with an increased fall risk in seniors.

Balance was also tested using a Freestep baropodometric platform (Sensor Medica, Rome, Italy). The platform consisted of a 40 cm × 40 cm active panel and a computer with dedicated software. During the test, the participants stood on the active panel. The following parameters were measured: surface of the ellipse, length of displacements and mean velocity of the center of pressure when standing on one leg. The test was performed for both the right and the left leg. The test lasted 10 s and began when a given participant lifted her leg up and assumed a relatively stable stance. If a participant was unable to stand for 10 s on one leg, the result was counted as a fall and discarded. The participants performed the test once on each leg. An assistant watched over the participants to ensure their safety [32].

The Biodex platform (Biosway, New York, NY, USA) is a device equipped with an appropriately configured platform and monitor. It also includes a foam covering to imitate unstable ground during one of the available tests. Biodex can help to assess balance, as well as improve it through the included training programs. The platform offers three standardized testing protocols and six interactive modes of training to match different problems and levels of fitness. The device provides a repeatable and objectively reliable assessment of neuromuscular control and balance on both stable and unstable ground. It can also help to evaluate treatment progress and document the rehabilitation of patients with balance disorders. The device is simple and convenient to use and is particularly recommended for assessing fall risk among seniors and in post-amputation rehabilitation, post-injury orthopedic rehabilitation, sports medicine programs, neuromuscular control disorders and screening before and after head injuries [33,34].

During the study on the Biodex platform, two test were performed: Limits of Stability (LoS)—this test assessed a participant’s ability to maintain his or her center of pressure outside the plane of support. The LOS for balance in a standing position were determined based on the maximum angle at which a participant was able to tilt away from the vertical position without losing balance. When the LOS were exceeded, the participant either fell or had to engage in corrective strategies to prevent a fall, such as taking a step with one leg or bending his or her knees [35]. The test involved mobilizing the participant, to move and control her center of pressure within the base of support, while keeping her feet on the ground. During each assessment, nine points were displayed on a screen. The participant was asked to look at the monitor and shift her body weight to make the cursor move from the center of the screen to a flashing point and back again as fast as possible.

Modified Clinical Test of Sensory Interaction on Balance (m-CTSIB)—the second test performed on the Biosway platform was the m-CTSIB, which assesses fall risk. The test is a reliable method for assessing balance disorders and can also identify disorders related to various systems engaged in postural control, i.e., visual control and the vestibular and somatosensory systems. The results of the test correlate strongly with fall risk [36,37]. In this study, the m-CTSIB was conducted once per participant. The test started after the participant took her position on the platform and was informed about the course of the test, and the positions of her feet were measured. The test consisted of four parts, each lasting 30 s. In the first part, participants stood on a solid surface to assess their visual, vestibular, and somatosensory control. In the second part, participants held a standing position with their eyes closed to assess their vestibular and somatosensory control. The third part involved holding a static position while standing on a dynamic surface with eyes open to assess the interaction between the visual and somatosensory systems. The fourth part was performed in the same manner as the third, except with eyes closed, to assess the interaction between the somatosensory system.

### 2.3. Therapeutic Intervention

The thirty women assigned to the experimental group took part in Pilates sessions. The training program lasted three months. Forty-five-minute sessions were held two times per week. Rhythmic music was played to make the sessions more attractive and make exercising easier. The calm songs, most of which the participants knew from their youth, had an additional soothing effect and helped them to focus on the exercises. This element of musical therapy was only secondary to the exercises, and was introduced to induce positive sensations.

The sessions were held in a large room with mirrors, which aided self-control and helped the participants to correct any mistakes. All exercises were explained to the participants in detail and demonstrated by the instructor beforehand. Each exercise was performed 10 times.

The Pilates sessions were divided into three parts:Introduction (approximately 10 min): warm-up, usually in a standing position, to help the participants assume the correct posture and prepare the body for the exercises, including exercises performed in a low position (described below). In addition to exercises that involved assuming and maintaining the correct posture, exercises involving balancing, strengthening or stabilizing the body were also conducted. In this part of the session, the participants learned the proper breathing method for Pilates. The following exercises were performed: pelvis rocking, toes standing, alternating knee lifting with foot point/flex exercises, rising arms up, spine twist and roll down/up.Main activity (approximately 30 min): exercises performed in different positions. The positions transitioned fluently from one to another through a slow rolling of the spine. This exercise is characteristic for Pilates and allows the trainee to transition from high positions to low positions and vice versa without rapid changes. It also improves elasticity, spine mobility and proprioception. Next, the participants performed exercises in low positions, i.e., while kneeling on both knees, kneeling on one knee, a four-point kneel and lying on the back, on the side and on the front. Participants also performed coordination and balance exercises. The following exercises were performed: superman, front support, swimming, the swan dive, single leg kick, the saw, the roll up, the hundred, the shoulder bridge, scissors, the one leg circle, arms circle and the side kick series.Cool-down (about 5 min): relaxation, stretching, and breathing exercises. The following exercises were performed: the one leg stretch, stretching of the gluteus medium muscles, stretching of the gastrocnemius muscles, body elongation, rest position, spine twist and spine stretch.

### 2.4. Statistical Methods

All obtained results were analyzed using the Statistica software, version 13.1 (StatSoft, Hamburg, Germany). Descriptive statistics concerning the studied parameters used the arithmetic mean, median, minimal and maximal values and standard deviation. 

Furthermore, the results were subjected to statistical analysis. The Shapiro–Wilk test was used to determine the normality of distribution in the tests, and Levene’s F test was used to test the homogeneity of variance. Both normal and abnormal distributions were observed. The significance of differences between the measurements was estimated using the Student–Goset *t*-test (in the samples with a normal distribution *p* > 0.05) or the Wilcoxon test (in the samples with an abnormal distribution *p* ≤ 0.05). To study the differences between groups, the t-test for independent samples (in samples with a normal distribution *p* > 0.05) and the Mann–Whitney U test (in samples with an abnormal distribution *p* ≤ 0.05) were used. For pairs of variables that differed statistically significantly, the effect size (Es-Standardized effect) was additionally estimated (Cohen’s D) [38].

## 3. Results

### 3.1. Timed Up and Go

The TUG test did not indicate any statistically significant differences in the experimental group following the Pilates program. However, the control group showed a statistically significant increase in TUG completion time by 9.4%. The value of the standardized effect indicates the great clinical significance of this change (Table 2).

### 3.2. One Leg Stance Test (OLST)

The OLST did not yield any statistically significant changes in the experimental group or the control group (Table 3).

### 3.3. Freestep Baropodometric Platform: Assessment of Balance during A One-Leg Stand

Statistical analysis was based on the results concerning the following parameters: surface of the ellipse, length of displacements and mean velocity at the center of pressure during a one-leg stand. Both legs were analyzed. The experimental group showed statistically significant differences in both the surface of the ellipse and the mean velocity at the center of pressure during a one-leg stand for the right leg: the former parameter decreased by 48.6% over the three months of Pilates training, and the latter improved by 17.2%. Regarding the significant differences, the effect size confirmed the significance of the change. For the other parameters, no statistically significant differences were found in the experimental or the control groups (Table 4).

### 3.4. Biosway Platform

#### 3.4.1. Limits of Stability

The LoS test on the Biosway platform yielded statistically significant changes in both the experimental and control groups. Over the three months, the mean score improved by 37.3% in the experimental group and by 14% in the control group compared to the first measurement. Regarding the significant differences, the effect size confirmed the significance of the change (Table 5).

#### 3.4.2. m-CTSIB

The results of m-CTSIB indicated a statistically significant difference in the experimental group when standing on an uneven surface with eyes closed. In this group, the mean score for this parameter improved by 2.3% over the three months of Pilates training. No significant differences were obtained in the other parts of the test, including standing on a solid surface with eyes open and closed, or standing on a soft surface with eyes open. The control group obtained no statistically significant results in any part of the test (Table 5).

## 4. Discussion

The aim of this study was to assess the effect of Pilates exercises on balance and fall risk in older women. The obtained results confirm the effectiveness of Pilates in improving balance and minimizing fall risk.

A study conducted by Pata et al. [39] demonstrated that Pilates training improved mobility and balance and decreased fall risk. Another study, conducted by Vieira et al. [40], also observed a positive result following a 12-week Pilates program. The results of the 6MWT (6-Minute Walk Test) and 5xSST (5 Times Sit to Stand Test) confirmed the beneficial effect of Pilates on dynamic balance, leg strength and fitness. However, the TUG and OLST yielded no statistically significant changes. In our study, no statistically significant improvement was also found in the experimental group. Furthermore, TUG results worsened in the control group. This confirms the key importance of regular physical activity in the form of Pilates training among seniors as a preventive measure against functional limitations and dependence on other people.

Even though the OLST is considered an effective research tool for assessing balance on a decreased plane of support, it is burdened with a risk of measurement error due to the human factor and non-uniform testing procedures [30]. This may lead to imprecise and subjective results. Consequently, this study also tested balance using a one-leg stand on a baropodometric platform, which allowed for more detailed and reliable results. The obtained results indicated a statistically significant improvement in the experimental group, who took part in Pilates training. For this group, the improvement concerned parameters such as the surface of the ellipse and mean velocity at the center of pressure in the right leg. Pilates training improved balance when lifting a leg for less than 10 s, which may have a considerable effect on everyday activities performed on a decreased plane of support, such as stepping into a bathtub, stepping over an obstacle or even walking and climbing stairs.

Key elements of maintaining balance include the appropriate reception of sensory stimuli coming in from the environment and the use of balancing strategies. The latter are responsible for maintaining balance when the center of pressure is located outside the plane of support [2]. Spink et al. [41] underlined the importance of the feet and the ankle joints in effectively maintaining balance. Their study, conducted in over 300 people aged over 65 years, demonstrated that the range of mobility in the ankle joint and the strength of the foot and hallux flexors were important determinants of balance and functional ability. This study used the LoS test performed on a balance platform to determine the effect of balancing strategies, especially the ankle strategy. The test was used to assess balance by determining the limits of stability in seniors following a Pilates training cycle. The participants in the experimental group improved their LoS by 37%, which was over twice as much as in the control group.

Balance disorders lead to falls, which are a particularly serious hazard to seniors. Therefore, physicians and physiotherapists should introduce preventive screening to limit the occurrence of falls. A good method for assessing balance and fall risk is the Romberg test and its modifications, i.e., the Sensory Organization Test (SOT) and the Clinical Test of Sensory Interaction on Balance (CTSIB), which are performed on special platforms designed to assess the effectiveness of various physical activities. To determine fall risk in seniors following Pilates training, Roller et al. [42] used a special SOT performed on a Neurocom platform. They also used other tests, such as TUG, the Berg Balance Scale (BBS), the 10 Meter Walk Test (10MWT) and the Activities-Specific Balance Confidence (ABC) scale. Pilates sessions were conducted on a reformer once per week for 10 weeks. The obtained results confirmed the effectiveness of Pilates in decreasing fall risk in seniors. Pyskir et al. [43], who assessed the effect of Pilates on postural stability in 35 older women, obtained slightly different results. Balance and fall risk were assessed using posturography after 10 weeks of 60 min classes held once per week. The results indicated no statistically significant differences in the improvement of stability following the Pilates training. Bergamin et al. [22] also observed no improvement in the parameters for balance, as assessed using the Romberg test on a balance platform with eyes open and closed. This study also performed tests on a balance platform to determine balance and fall risk using a modified CTSIB, composed of four parts. The first two parts were identical to the test performed by Bergamin et al. [22] and, likewise, no statistically significant improvement was observed. However, this study did obtain a significant improvement in the part of the test conducted on a soft surface with eyes closed, which Bergamin et al. [22] did not include in their assessment. The positive result of the test performed on a soft surface may indicate that Pilates is in fact an effective method for improving proprioception and spatial awareness. Improved proprioception and the proper reception of stimuli from surroundings are crucial for good balance. Efficient control over one’s body greatly minimizes the risk of falls, which in seniors are often tied to proprioceptive disorders that progress with age.

Summing up, imbalance and fall risk are very important issues in an aging society. In our own research, Pilates exercises improved balance by increasing the limits of stability. There was also an improvement in balance on an unstable surface during the m-CTSIB test and in the one-leg stand test on the Freestep platform. Improvement of these parameters is important in the better performing of daily activities and reducing fall risk.

Additionally, Pilates seems to be a good alternative to tedious and often less attractive exercises, but in the current literature there is no unequivocal evidence for this. The authors only agree that any kind of physical training has a positive impact on balance [44,45,46]. From the analyzed studies, it is unclear whether one specific training regimen may be more beneficial than others in order to increase balance in the elderly. However, all the no-activity groups showed a decline in function.

## 5. Study Limitations

This study is not without limitations. A larger sample and the addition of a control group would be good guidelines for future studies. Furthermore, all participants of this study lived in a large city, meaning that women from small cities or villages were not included. This is important due to the fact that a rural lifestyle is slightly different to an urban lifestyle, and that a larger-scale study could bring a broader insight into the subject matter. Sample size was also technically and organizationally limited by: access to people who responded positively to the invitation to participate in the research and met the inclusion criteria; limited funds that the authors of the project could allocate for research (the project was entirely financed from the authors’ own funds); and lack of literature data that would make it possible to calculate the required sample size at the research planning stage.

## 6. Conclusions

Although no statistically significant differences were observed between groups, more balance indicators significantly improved in the group with Pilates training (the experimental group). Pilates training influenced the balance of the examined women by significantly increasing the surface of the ellipse, mean velocity and limits of stability. Pilates training reduced fall risk in the examined women by significantly increasing the results of the m-CTSIB with eyes closed on an unstable surface. Reduced fall risk is very important in everyday functioning because it gives a sense of stability and movement freedom.

## Figures and Tables

**Figure 1 ijerph-18-03663-f001:**
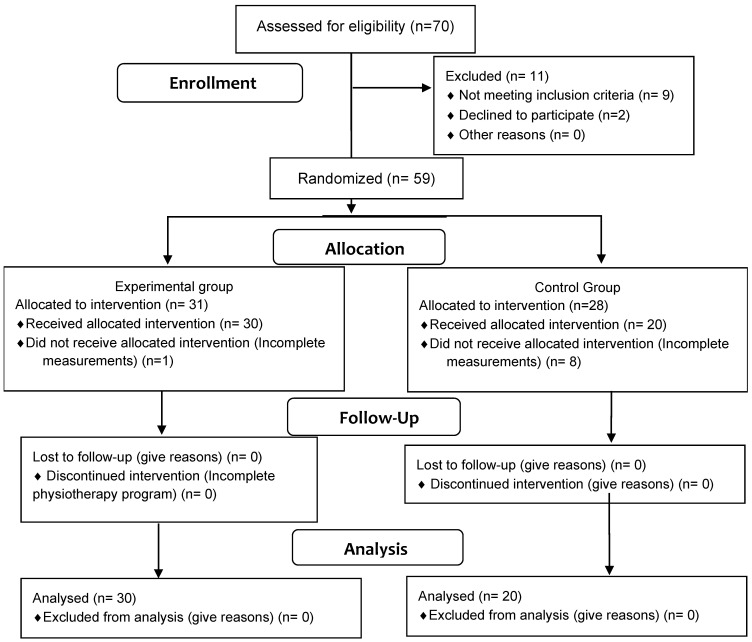
Consort diagram.

**Table 1 ijerph-18-03663-t001:** Sample characteristics.

Parameter	Experimental Group						Control Group						
	*N*	$\bar{x}$	Me	SD	Min	Max	*N*	$\bar{x}$	Me	SD	Min	Max	*p*
Age (years)	30	67.73	67.00	4.10	61.00	77.00	20	68.10	68.00	3.35	62.00	76.00	0.741
Height (m)	30	1.58	1.58	0.05	1.45	1.66	20	1.60	1.60	0.06	1.47	1.70	0.134
Weight (kg)	30	70.72	72.95	9.45	54.00	89.70	20	71.34	71.40	12.71	52.00	110.5	0.845
BMI	30	28.45	28.34	3.78	22.33	38.32	5	27.73	27.84	4.11	21.93	40.60	0.539

Me—median.

**Table 2 ijerph-18-03663-t002:** Timed Up and Go (TUG) results in the experimental and control groups.

	Experimental Group					Control Group					Between Groups Comparison
	$\bar{x}$	SD	Me	Min	Max	$\bar{x}$	SD	Me	Min	Max	
pretest	8.41	1.50	8.34	5.00	12.65	8.71	1.43	8.69	6.59	11.65	*p* = 0.496 Es = 0.03
posttest	8.84	1.63	8.66	6.31	13.57	9.53	1.04	9.58	7.12	11.28	*p* = 0.097 Es = 0.07
Pre to post-test comparison	*p* = 0.148, Es = 0.27					*p* = 0.009 *, Es = 0.64					

Me—median; Es—standarized effect; * significantly different than the respective value taken before the training (*p* < 0.05).

**Table 3 ijerph-18-03663-t003:** One Leg Stance Test (OLST) results for the experimental and control groups.

		Experimental Group					Control Group					Between Groups Comparison
		$\bar{x}$	SD	Me	Min	Max	$\bar{x}$	SD	Me	Min	Max	
Right leg	pretest	22.67	9.85	30.00	2.69	30.00	22.51	10.35	30.00	5.00	30.00	*p* = 0.835 Es = 0.01
	posttest	22.54	10.18	29.55	3.00	30.00	20.62	10.49	24.40	1.31	30.00	*p* = 0.620 ES = 0.20
Pre- to post-test comparison		*p* = 0.879, Es = 0.01					*p* = 0.424, Es = 0.23					
Left leg	pretest	17.82	10.93	17.00	1.00	30.00	19.87	10.84	22.00	1.84	30.00	*p* = 0.458 Es = 0.21
	posttest	21.60	8.62	21.88	4.66	30.00	18.62	10.11	20.11	1.00	30.00	*p* = 0.941 Es = 0.30
Pre- to post-test comparison		*p* = 0.078, Es = 0.31					*p* = 0.653, Es = 0.09					
Dominant leg	pretest	21.62	10.30	27.50	2.69	30.00	21.40	9.96	26.50	3.34	30.00	*p* = 0.965 Es = 0.02
	posttest	23.09	9.40	29.55	3.00	30.00	20.85	9.48	22.73	1.31	30.00	*p* = 0.772 Es = 0.22
Pre- to post-test comparison		*p* = 0.506, Es = 0.12					*p* = 0.778, Es = 0.06					

Me—median; Es—standarized effect.

**Table 4 ijerph-18-03663-t004:** Results of balance assessment during one-leg stand on a Freestep platform.

			Experimental Group					Control Group					Between Groups Comparison
			$\bar{x}$	SD	Me	Min	Max	$\bar{x}$	SD	Me	Min	Max	
Right leg	Surface of ellipse (cm)	pretest	1749.45	2235.25	946.20	127.80	9145.91	3456.01	3634.55	1593.95	256.39	8888.10	*p* = 0.098 Es = 0.48
		posttest	567.42	370.54	486.01	198.30	1592.88	1556.95	2327.15	629.74	195.09	8443.99	*p* = 0.135 Es = 0.44
	Pre- to post-test comparison		*p* = 0.004 *, Es = 0.54					*p* = 0.155, Es = 0.72					
	Length of displacement (mm)	pretest	596.26	207.40	531.82	371.13	1148.00	603.79	254.32	512.09	349.50	1008.91	*p* = 0.945 Es = 0.75
		posttest	490.62	89.93	486.39	310.95	678.36	501.89	111.85	495.75	312.51	655.40	*p* = 0.642 Es = 0.52
	Pre- to post-test comparison		*p* = 0.057, Es = 47					*p* = 0.248, Es = 0.38					
	Mean velocity (mm/s)	pretest	48.84	19.91	43.02	27.19	101.64	49.99	24.21	40.99	21.07	89.36	*p* = 0.051 Es = 0.16
		posttest	37.54	8.10	35.62	22.97	53.18	38.64	10.61	37.94	20.14	57.13	*p* = 0.255 Es = 0.11
	Pre- to post-test comparison		*p* = 0.026 *, Es = 0.67					*p* = 0.109, Es = 0.46					
Left leg	Surface of ellipse (cm)	pretest	971.55	1251.66	700.14	246.43	6262.33	1000.15	1258.69	677.46	260.53	5141.94	*p* = 0.992 Es = 0.11
		posttest	702.30	654.86	564.43	179.55	3350.45	1293.56	1752.40	818.89	244.67	7098.40	*p* = 0.389 Es = 0.29
	Pre- to post-test comparison		*p* = 0.263, Es = 30					*p* = 0.826, Es = 0.56					
	Length of displacement (mm)	pretest	545.28	106.83	534.00	364.83	909.96	576.07	190.12	563.14	280.27	1134.69	*p* = 0.2544 Es = 0.43
		posttest	556.81	143.62	522.77	415.66	1105.05	583.58	141.39	562.46	336.47	837.50	*p* = 0.7892 Es = 0.08
	Pre- to post-test comparison		*p* = 0.592, Es = 0.08					*p* = 0.551, Es = 0.03					
	Mean velocity (mm/s)	pretest	40.41	13.37	37.75	21.88	87.39	44.23	21.71	43.31	22.24	112.19	*p* = 0.110 Es = 0.38
		posttest	36.77	13.28	33.67	22.67	90.39	42.00	16.11	34.37	25.19	77.53	*p* = 0.759 Es = 0.52
	Pre- to post-test comparison		*p* = 0.072, Es = 0.20					*p* = 0.875, Es = 0.09					

Me—median; Es—standarized effect; * significantly different than the respective value taken before the training (*p* < 0.05).

**Table 5 ijerph-18-03663-t005:** Limits of Stability (LoS) and Modified Clinical Test of Sensory Interaction on Balance (m-CTSIB) results.

		Experimental Group					Control Group					Between Groups Comparison
		$\bar{x}$	SD	Me	Min	Max	$\bar{x}$	SD	Me	Min	Max	
Limits of Stability	pretest	27.850	9.366	28.000	9.000	45.000	30.300	9.398	29.000	13.000	44.000	*p* = 0.738 Es = 0.24
	posttest	38.250	13.981	36.500	14.000	65.000	34.550	9.378	36.000	16.000	49.000	*p* = 0.675 Es = 0.37
Pre to post-test comparison		*p* = 0.005 *, Es = 0.68					*p* = 0.015 *, Es = 0.59					
m-CTSIB eyes open stable surface	pretest	0.775	0.316	0.710	0.270	1.750	0.763	0.278	0.675	0.430	1.590	*p* = 0.866 Es = 0.01
	posttest	0.772	0.313	0.720	0.420	2.120	0.662	0.233	0.645	0.330	1.240	*p* = 0.127 Es = 0.01
Pre to post-test comparison		*p* = 0.674, Es = 0.02					*p* = 0.117, Es = 0.36					
m-CTSIB eyes closed stable surface	pretest	0.742	0.286	0.700	0.400	1.710	0.817	0.327	0.710	0.460	1.740	*p* = 0.411 Es = 0.01
	posttest	0.688	0.230	0.675	0.390	1.320	0.744	0.372	0.590	0.370	1.580	*p* = 0.759 Es = 0.01
Pre to post-test comparison		*p* = 0.245, Es = 0.21					*p* = 0.104, Es = 0.30					
m-CTSIB eyes open unstable surface	pretest	1.805	0.321	1.855	1.260	2.440	1.769	0.289	1.795	1.380	2.360	*p* = 0.649 Es = 0.01
	posttest	1.780	0.407	1.865	0.970	2.860	1.665	0.233	1.725	1.230	1.970	*p* = 0.195 Es = 0.01
Pre to post-test comparison		*p* = 0.690, Es = 0.07					*p* = 0.107, Es = 0.37					
m-CTSIB eyes closed unstable surface	pretest	2.936	0.620	2.750	1.770	4.820	2.791	0.614	2.640	2.050	4.570	*p* = 0.223 Es = 0.01
	posttest	2.752	0.784	2.640	1.590	4.820	2.492	0.550	2.530	1.620	3.880	*p* = 0.216 Es = 0.04
Pre to post-test comparison		*p* = 0.041 *, Es = 0.18					*p* = 0.065, Es = 0.42					

Me—median; Es—standarized effect; * significantly different than the respective value taken before the training (*p* < 0.05).

## Data Availability

The data presented in this study are available on request from the corresponding author.

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
