# Peer review of "Effect of Three Months Pilates Training on Balance and Fall Risk in Older Women"

_ijerph, 2021, doi:10.3390/ijerph18073663_

Round 1

Reviewer 1 Report

Congratulations. The manuscript has significantly improved after all major changes.

Author Response

Dear Reviewer,

We would like to thank you for accepting our manuscript.

Reviewer 2 Report

I have only few technical comments for this re-submitting manuscript:

Line 239: please remove the dot

Table 3: remove the dots from some of the values 

Tables 1,2,3: The term "intra" means IN group, "inter" means between the groups

My suggestion for the columnes title is: Inter group comparison. Then, for the line in the Tables: Intra (pre vs. post intervention) comparison.

Author Response

Dear Reviewer,

We would like to thank you for a detailed review of our manuscript and all of your valuable remarks. But we feel that Reviewer had a previous version of our manuscript, because in current version (ijerph-1160043-ori) we can`t find any of Reviewer`s remarks. In current version of our manuscript all remarks were considered earlier. 

  1. In line 239 we can only found a dot which finish the sentence. There is no other dots we can remove.
  2. We can not find any extra dots or double dots in Table 3, so we don`t know which dots we have to remove.
  3. There is no “intra” or “inter” words in Table 1, 2, and 3.
  4. There is no “inter group comparison” column in Tables.

Reviewer 3 Report

The presentation of the results in the revised manuscripts is pool. Also, the conclusion did not support the experiment design and results. Several minor problems, including style, format, and typos should be checked again.

  1. It is hard to track the revision parts based on the author’s response note. The authors should clearly describe that the revisions have been done in the revised manuscript. For example, the authors claimed that appropriate references are updated in the revised manuscript. The authors should clearly list the updated referenced in the note and describe their importance. Also, the exactly updated positions of the revisions should be provided for the reviewers.
  2. There are several typos and format problems in Table 2-5.
    1. Please define intra-group and inter-groups.
    2. What is “Intra Between groups comparison”?
    3. What is “Inter grouppre to posttest comparison?
    4. The unit should be added for parameters. For example, the unit of “mean velocity” might be (m/s^2).
    5. The comma should be added to separate “p=0.0037* Es=0.54”, which can be modified as “p=0.0037*, Es=0.54”. A similiar modification should be updated to other values.
    6. Please describe the statistical meaning of Es(Standardized effect) in Section 2.5 “Statistical Methods”.
  3. In Conclusion, the authors claimed ”Therefore, this form of exercise, under the supervision of qualified specialists, should be recommended to older women, especially to women with a sedentary lifestyle.
    1. How did this work define sedentary lifestyle?
    2. Did this study recruit the subjects with sedentary lifestyle? If yes, please modify the experiments.
    3. This work did not design the experimental results that can support such a conclusion.

Author Response

Dear Reviewer,

We would like to thank you for a detailed review of our manuscript and all of your valuable remarks. We have addressed them all in detail below.

1 - All changes in our manuscript we marked by „Track changes” due to Editor`s order

2 - (1 -3) - In current version of our manuscript (ijerph-1160043-ori) there is no such format problems in Tables 2-5

4 - We added units for parameters due to Reviewer`s suggestion.

5 - We added comas due to Reviewer`s suggestion

6 - This meaning is described in the last sentence of ‘Statistical method” section. We only filled this sentence.

3 (1-3) – we agree that this sentence has no strong connetions with our study, but previous Reviewers gave us a suggestions to put it into conclusion section. But we removed this sentence due to present Reviewer`s suggestion.

Round 2

Reviewer 3 Report

The authors have fully revised the paper. The reviewer suggests accepting the paper for publication. 

This manuscript is a resubmission of an earlier submission. The following is a list of the peer review reports and author responses from that submission.

Round 1

Reviewer 1 Report

General comment: the topic of the manuscript is very interesting but several major changes need to be made.

Abstract:

  1. Pg. 1 – L13: Please, edit: “experimental group (n = 30), which took part in Pilates sessions twice a week for three months, and control group (n = 20)”.

Introduction:

  1. There are some paragraphs which consist of only one sentence. Please, edit and try to keep a link between sentences/paragraphs in the Introduction.
  2. There are some references cited in two different styles (e.g., L34, L39, L43, etc.). Please, check all the manuscript and correct following the journal’s style.
  3. Please, add reference: “Maintaining functional fitness and, consequently, independence seems to be the key factor affecting the psychomotor condition of seniors. Balance disorders and falls are two significant issues that may impact the functioning of an aging population”.
  4. Please, add reference “Involutionary changes related to aging affect all primary systems of the body, including anatomical and functional systems”.
  5. Please, add references “The main task of the balance system is to keep the body’s center of mass at equilibrium, both at rest and during activity, by generating the appropriate responses to control 36 body posture. These responses are based on information coming from the proprioceptors, eyes, and vestibular system”.
  6. Please, add references: “Furthermore, neural conductivity diminishes, leading to longer reaction times, and the integration of sensory and mo-44 tor responses becomes impaired”.
  7. Regarding the importance of physical activity for balance and functional performance of the body, I guess the authors should talk about the importance of activating the core muscles of the body through Pilates exercises. I suggest checking this reference: Int. J. Environ. Res. Public Health 2020, 17(12), 4306; https://doi.org/10.3390/ijerph17124306
  8. Please, add references: “An important factor in maintaining balance in seniors is cultivating appropriate muscle mass and muscle strength. Proper conditioning, especially of the legs, may prevent loss of balance”.

Materials and methods:

  1. Table 1 is empty. Please, complete. Also, I suggest changing the title: “Sample characteristics”.
  2. Regarding the exclusion criteria, please remove the bullet points.
  3. Pg. 3 – L95. Please, correct “poor mental health (e.g., depression”).
  4. Pg. 3 – L105. Please, correct “training cycle (i.e., three months later)”.
  5. Pg. 3. Please, remove all bullet points in the manuscript.
  6. Why did you choose these tests? Add references that justify why these tests may be used for your outcome variables.
  7. Pg. 3 – L108: The authors mention that TUG test was used to assess gait and functional state while in L119 it is mentioned that TUG test assesses functional fitness and fall risk. Please, correct based on the second sentence since it is the most accurate regarding the aims of this study. Indeed, it would be better if the authors mix the information from L108 to L113 with the information from 2.3.1, 2.3.2, and 2.3.3. Therefore, it would not be necessary to create “2.3. Measurement tools, 2.3.1. etc.”. Then, the paragraph would be something like this: “Body mass was measured using a Tanita scale (Company, City, Country) to an accuracy of 100 g and body height was measured using a Martin anthropometer to an accuracy of 1 mm (Company, City, Country). Specifically, these variables were measured once. In addition, regarding the balance and fall risk, these were measured twice. In this regard, Timed Up and Go (TUG) was used to assess functional fitness and fall risk [22]…”. In summary, the explanation of the methods section needs to be specific and link variable of analysis and the instrument used to measure such variable. Perhaps, some information needs to be removed regarding the description of each test or instrument. For instance, “Timed Up and Go is a very simple and quick test”.
  8. “The average score for persons over 60 years is assumed to be 8 seconds [23]”. I guess this sentence may be better in the discussion.
  9. Sensor Medica baropodometric platform. Please, add (Company, City, Country). This is also required for Biodex BIOSWAY system.
  10. Pg. 5 – L214: Please, replace main “portion” by “main activity”.
  11. Pilates intervention: please, this is one of the most important parts of the manuscript. The reader needs to know exactly which exercises were done, how many sets/repetitions, duration, etc.
  12. Add (Company, City, Country) for Statistica software.

Results:

  1. The effect size for each comparison needs to be added.
  2. Tables: what does “Me” mean? Please, explain in the table notes. Also, move SD column next to the mean.
  3. Tables: try to keep all the words within the same row. For instance: surface of ellipse is in two rows. Please, try to keep the same row or do not “cut” the words between rows.

Discussion:

  1. Please, add limitations of the study and make proposals for future studies.

Conclusion:

  1. Please, rewrite. The reader should have a clear “take-home message”. “Pilates training had a significant effect on balance…”. Also, try to add some practical applications of the study.

Author Response

Reviewer 1

Dear Reviewer,

We would like to thank you for a detailed review of our manuscript and all of your valuable remarks. We have addressed them all in detail below.

Abtract:

  1. We edited sentence due to Reviewer’s suggestion.

Introduction:

  1. We slightly rebuilt Introduction due to Reviewer’s suggestion.
  2. We corrected style of references citation according to journal style in whole manuscript due to Reviewer’s suggestion.
  3. – 7. We added references due to Reviewer’s suggestion
  4. We cited reference due to Reviewer’s suggestion
  5. We removed this sentence during rebuilding process.

Materials and methods:

  1. We completed Table 1 and changed the table title due to Reviewer’s suggestion
  2. and 14. We removed all bullet points in whole manuscript due to Reviewer’s suggestion
  3. and 13. We edited sentences due to Reviewer’s suggestion
  4. We decided to choose such testes because – according to literature – those tests are commonly used in research on older people as a safe, easy to understand for patient, reliable and repetitive tools.
  5. We standardized TUG description due to Reviewer’s suggestion. We completed information about Tanita and anthropometer due to Reviewer’s suggestion. We removed sentence due to Reviewer’s suggestion. We rebuilt whole paragraph (Methods) due to Reviewer’s suggestion.
  6. We removed sentence due to Reviewer’s suggestion.
  7. We completed information about Biosway and FreeMed platforms due to Reviewer’s suggestion
  8. We replaced word due to Reviewer’s suggestion
  9. We completed information about Pilates intervention due to Reviewer’s suggestion
  10. We completed information about Statistica due to Reviewer’s suggestion

Results:

  1. We added effect size for each comparison due to Reviewer’s suggestion
  2. And 24. We explained abbreviation „Me” in the Table notes and rebuilt all tables due to Reviewer’s suggestion

Discussion:

  1. We added „Study limitation” paragraph due to Reviewer’s suggestion

Conclusion:

  1. We corrected conclusions due to Reviewer’s suggestion

Reviewer 2 Report

The Authors of the manuscript entitled “Effect of Pilates Exercises on Balance and Fall Risk in Elderly Women” aims to main goal of this study was to compare the impact of Pilates exercise on Balance and fall risk using TUG, OLST, Baropodometric platform and Biosway platform.

The results of the study show that Pilates exercises improved significantly some parameters from baropodometric platform in experimental group and from Biosway Platfrom in experimental and control group

Pilates exercise for older people is very interesting theme for researchers but there is a lot of major comments after review this manuscript:

  • There is no information about Clinical Trial Registration
  • How Authors conducted their study? Why Consort statement did not use?
  • No differences were presented between the groups, why? In my opinion you must improve your statistics to show intra and intergroup comparison
  • In relation to the methodology: Why was the sample size not calculated? Why you did not planned the follow-up strategy? How was randomization performed?
  • Can the authors demonstrate the Pilates Intervention? In my opinion it is necessary for reply the intervention
  • Discussion section should be slightly summarized. However it should be more discussed the differences were found between the groups
  • In my opinion the conclusion does not provide sufficient information (because of the lack of information from inter group comparison)
  • What is the study limitations?
  • In my opinion we should try to avoid language that might be deemed unacceptable or inappropriate – “older people” is preferred to 'the elderly'
  • I read carefully the references and I think that they could be more newest 

Author Response

Reviewer 2

Dear Reviewer,

We would like to thank you for a detailed review of our manuscript and all of your valuable remarks. We have addressed them all in detail below.

  1. We did`t add the information about Clinical Trial Registration because IJERPH doesn`t required such information.
  2. We added Consort diagram due to Reviewer’s suggestion
  3. We added inter and intra group comparisons due to Reviewer’s suggestion
  4. We added in Method section information about randomization due to Reviewer’s suggestion. We want to explain that sample size was technically and organizationally limited: access to people who have responded positively to the invitation to participate in the research and met the inclusion criteria; limited funds that the authors of the project could allocate for research (the project was entirely financed from the authors' own funds); lack of literature data that would make it possible to calculate the required sample size at the research planning stage.
  5. We completed information about Pilates intervention due to Reviewer’s suggestion
  6. We summarized Discussion section due to Reviewer’s suggestion. We also tried to describe the research results in detail and refer to the literature.
  7. We corrected conclusions due to Reviewer’s suggestion
  8. We added „Study limitation” paragraph due to Reviewer’s suggestion.
  9. We corrected “older people” to “the elderly” in whole manuscript due to Reviewer’s suggestion
  10. We added some new publications in References section due to Reviewer’s suggestion.

Reviewer 3 Report

Generally, the paper is well writing. The organization is well. However, the more related should be surveyed. The minor comments are listed as follows:

  1. Please survey the effects  of other exercisess for balance assessment in the elderly.
  2. There are so many exercises have the potential to improve the balance. Why did this work only choose Pilates? Please compare with other exercisess.
  3. Is it possible to divide the elderly into different groups according to their balance status?

Author Response

Reviewer 3

Dear Reviewer,

We would like to thank you for a detailed review of our manuscript and all of your valuable remarks. We have addressed them all in detail below.

  1. We added appropriate references due to Reviewer’s suggestion.
  2. We added appropriate references and we explained our choice (Pilates) due to Reviewer’s suggestion
  3. There is possibility to divide patients into groups according to their balance status but we decided not to do it because we wanted to maintain clear table form.

Round 2

Reviewer 1 Report

Overall, the manuscript has improved but some changes are required. In addition, English proofreading is needed before publication.

Introduction:

  1. Please, add reference: “Maintaining functional fitness and, consequently, independence seems to be the key factor affecting the psychomotor condition of seniors
  2. Please, add reference “Involutionary changes related to aging affect all primary systems of the body, including anatomical and functional systems”.
  3. Please, add references “The main task of the balance system is to keep the body’s center of mass at equilibrium, both at rest and during activity, by generating the appropriate responses to control body posture.

Materials and methods:

  1. Please, use dots (.) instead of o commas (,) in the Tables.
  2. I do not understand why the authors mention that “All participants of this study performed two tests each, one at the beginning and one at the end of the study” once they have already mentioned that other tests were performed before. Please, be clear in this part of the manuscript. The reader needs to have a clear idea of the variables that you are testing and the tests/instruments that you use to measure these variables. Please, change this part of the manuscript.
  3. Add (Company, City, Country) for Statistica software. This was not changed correctly. StatSoft is set in Hamburg (Germany). Then it might be: (StatSoft, Hamburg, Germany).

Results:

  1. Please, change these sentences: “For statistically significant results…” by “Regarding the significant differences…”. There are several sentences like this one. Also, avoid closing the sentence with dot “.” if parenthesis (Table 5) is after it. Just close the sentence after the end of the parenthesis.

Discussion:

  1. “Oliveira et 619 al. [44] and Josephs et al. [45] shown”. Please, replace “shown” by “showed”.

Author Response

Reviewer 1

Dear Reviewer,

We would like to thank you for a detailed review of our manuscript and all of your valuable remarks. We have addressed them all in detail below.

Introduction:

  1. – We added references due to Reviewer’s suggestion

Materials and methods:

  1. We changed comas to dots in all tables due to Reviewer’s suggestion
  2. We removed this sentence because it introduced unnecessary confusion. This sentence was a repeating of the sentence form the beginning of this section.
  3. We added correct set of Statistica due to Reviewer’s suggestion

Results

  1. We changed all mentioned sentences and corrected dots in sentences due to Reviewer’s suggestion

Discussion

  1. We corrected word due to Reviewer’s suggestion

Reviewer 2 Report

  1. We did`nt add the information about Clinical Trial Registration because IJERPH doesn`t required such information: Ok, but in my opinion Trial Registration is necessary.
  2. We added Consort diagram due to Reviewer’s suggestion. Thank you, do you have also Consort checklist in your project's documents?
  3. We added inter and intra group comparisons due to Reviewer’s suggestion. Indeed, You added intra group comparisons, but for me the Tables are not clear enough...which statistics value comes from intra group comparisons and which comes from inter group comparisons? The whole re-submitted manuscript is not clear.
  4. We added in Method section information about randomization due to Reviewer’s suggestion. We want to explain that sample size was technically and organizationally limited: access to people who have responded positively to the invitation to participate in the research and met the inclusion criteria; limited funds that the authors of the project could allocate for research (the project was entirely financed from the authors' own funds); lack of literature data that would make it possible to calculate the required sample size at the research planning stage. Ok, please add this information to the study limitation.
  5. We completed information about Pilates intervention due to Reviewer’s suggestion. Thank you.
  6. We summarized Discussion section due to Reviewer’s suggestion. We also tried to describe the research results in detail and refer to the literature. I still cannot find in the discussion: Generalisability (external validity, applicability) of the trial findings and Interpretation consistent with results, balancing benefits and harms, and considering other relevant evidence.

Line 547-555 needs a refference.

Line 563-564 The Authors write about LOS in the experimental group, but there is no difference between the groups. Then, Authors write  "the trainees felt like they were able to control their stability (...) Which affects everyday activities by improving psychomotor function" and it sounds great, but where is the evidence? Did the Authors compare the LOS from Biodex with daily activities questionnaires?

7. We corrected conclusions due to Reviewer’s suggestion. In my opinion the conclusion is still unclear, especially because there is no statistics differences between the groups. Am I right?

8. We added „Study limitation” paragraph due to Reviewer’s suggestion. OK

9. We corrected “older people” to “the elderly” in whole manuscript due to Reviewer’s suggestion. OK

10. We added some new publications in References section due to Reviewer’s suggestion. OK

Author Response

Reviewer 2

Dear Reviewer,

We would like to thank you for a detailed review of our manuscript and all of your valuable remarks. We have addressed them all in detail below.

  1. Unfortunately we didn`t register this project, so we can`t add such information .
  2. Yes, we have Consort checklist in our project`s documents.
  3. We have added an explanation in the tables which statistical values are inter and intra group comparisons due to Reviewer`s suggestions. The large amount of data in the table are connected with expectations of two Reviewers (we had to add calculations inter and intra group comparisons and Standarized effect values). We tried to make the table layout consistent with the format preferred by most scientific journals. We would like to emphasize that the present shape of the manuscript is the result of a compromise between the expectations of the three reviewers, and we wanted to apply them to their comments.
  4. We added information about sample size in the Study limitation section due to Reviewer’s suggestion
  5. Thank you for accepting our corrections
  6. We would like to point out that our discussion is conducted in accordance with the guidelines for writing this part of the scientific article.We tried our best to compare the results of our research with the reports of other authors.In order not to reduce the readability of this part of the work, at the end of this chapter, we have included – due to Reviewer’s suggestion - a summary and a paragraph describing the advantages of the Pilates method in relation to other training methods in older people, along with emphasizing the lack of compliance in this regard among scientists.

We added references due to Reviewer’s suggestion.

We removes mentioned sentences because in fact although they are true, they do not result from our research.

  1. We completed conclusions due to Reviewer’s suggestion.
  2. Thank you for accepting our corrections.
  3. Thank you for accepting our corrections
  4. Thank you for accepting our corrections